# Iron Deficiency in Pulmonary Arterial Hypertension: A Deep Dive into the Mechanisms

**DOI:** 10.3390/cells10020477

**Published:** 2021-02-23

**Authors:** Marceau Quatredeniers, Pedro Mendes-Ferreira, Diana Santos-Ribeiro, Morad K. Nakhleh, Maria-Rosa Ghigna, Sylvia Cohen-Kaminsky, Frédéric Perros

**Affiliations:** 1INSERM UMR_S 999, Hypertension Pulmonaire: Physiopathologie et Innovation Thérapeutique, Hôpital Marie Lannelongue, F-92350 Le Plessis-Robinson, France; quatredeniers.marceau@gmail.com (M.Q.); mnakhleh@gmail.com (M.K.N.); mrghigna@gmail.com (M.-R.G.); sylvia.cohen-kaminsky@u-psud.fr (S.C.-K.); 2Faculté de Médecine, Université Paris-Saclay, F-94270 Le Kremlin-Bicêtre, France; 3Cardiovascular R&D Center, Faculty of Medicine of the University of Porto, 4200-450 Porto, Portugal; pmferreira@med.up.pt; 4Paris-Porto Pulmonary Hypertension Collaborative Laboratory (3PH), UMR_S 999, INSERM, Université Paris-Saclay, F-94270 Le Kremlin-Bicêtre, France; 5Institute of Experimental and Clinical Research (IREC), Pneumology, ENT and Dermatology, Université catholique de Louvain (UCL), 1060 Brussels, Belgium; diana.santos@uclouvain.be

**Keywords:** pulmonary arterial hypertension, iron deficiency, cardiovascular disease, iron homeostasis

## Abstract

Pulmonary arterial hypertension (PAH) is a severe cardiovascular disease that is caused by the progressive occlusion of the distal pulmonary arteries, eventually leading to right heart failure and death. Almost 40% of patients with PAH are iron deficient. Although widely studied, the mechanisms linking between PAH and iron deficiency remain unclear. Here we review the mechanisms regulating iron homeostasis and the preclinical and clinical data available on iron deficiency in PAH. Then we discuss the potential implications of iron deficiency on the development and management of PAH.

## 1. Introduction

Iron deficiency is the most widespread nutritional deficiency worldwide and the 13th most serious when measured by disability-adjusted life year (DALY) (a measure of the number of years lost due to illness) [1]. Iron deficiency can lead to anemia, which affects more than 45 million people worldwide, mainly children and women [1,2]. Iron deficiency affects less than 20% of the population in developed countries, where it is related to iron poor diets (e.g., vegetarianism and veganism), chronic infections, chronic blood loss, and doping (e.g., recombinant human erythropoietin injections), while it affects 30–70% of the population in developing countries, mainly due to low iron diets and parasitic infections [2,3]. Globally, approximately 2 billion people have iron deficiency [4].

Iron (Fe) is required for vital functions, including oxygen transport, DNA synthesis, and cellular metabolism [5,6]. On the other hand, it can also result in the production of reactive oxygen species through the Fenton reaction, leading to oxidative stress and, subsequently, lipid peroxidation, DNA instability, and replication defects responsible for cellular deterioration [5,6,7,8]. Therefore, tight regulation of iron metabolism is required to prevent iron deficiency or overload, both of which can be detrimental. Iron also plays important roles in numerous physiological processes, including pregnancy, hypoxia, erythropoiesis, inflammation, and menstruation [9]. Consequently, while iron overload is implicated in hemochromatosis and sometimes chronic blood transfusion, iron deficiency is observed in parasitic infections, cancers, anemia, neurodegenerative diseases, some genetic disorders, and cardiovascular diseases (such as pulmonary arterial hypertension) [9].

Pulmonary arterial hypertension (PAH) is a severe form of pulmonary hypertension (PH) with decreased serum iron levels [10,11,12]. However, whether iron deficiency contributes to or is merely a consequence of the disease remains debated. The present review aims to provide an overview of the mechanisms regulating iron homeostasis for a general audience before focusing on the manifestation of iron deficiency in PAH and experimental models, the current molecular evidence linking iron dyshomeostasis to PAH, and its clinical implications and management. In a perspective manner, the potential of iron supplementation in disease management is explored.

## 2. Iron Homeostasis: Mechanisms of Regulation

Healthy individuals have around 3–4 g of iron, which is tightly regulated [8]. The gut absorbs approximately 1–2 mg of iron daily, and the same amount is lost mainly through desquamation in the skin, gut, and urinary tract; sweating; and occasionally, blood loss [13]. Since both iron overload and deficiency are deleterious, a tight regulation of iron balance is essential—from uptake in the gut to storage in the blood and liver (Figure 1). These mechanisms, and the contribution of the various organs to iron homeostasis, are described below.

### 2.1. Gut

Iron uptake occurs in the gut, with most of the absorption taking place in the duodenum. Diet-derived ferric iron (Fe^3+^) is reduced at the apical surface of enterocytes by the duodenal cytochrome b ferrireductase (Dcytb) to the more labile ferrous iron (Fe^2+^), which is transported into the enterocytes by the divalent metal transporter 1 (DMT-1) [14,15]. Iron may also be bound to heme-containing proteins, mainly hemoglobin, in which case heme-Fe is transported through the membrane-anchored enterocytic heme carrier protein 1 (HCP1, also known as proton-coupled folate transporter or SLC46A1) [16]. Once internalized in lysosomes, iron is released from heme by the inducible heme oxygenase 1 (HO-1). Fe^2+^ is then stored within the cytoplasm by binding to ferritin, a large globular protein that can bind up to 4500 iron atoms in its central cavity [17]. Fe^2+^ is exported from the enterocytes to the circulatory system by ferroportin (FPN-1) [18,19]. At the cell surface, Fe^2+^ is oxidized by hephaestin (HEPH) to Fe^3+^, which then binds to transferrin (Tf) in the plasma, allowing its delivery to the organs [20].

### 2.2. Liver

The liver is one of the main tissues dedicated to iron storage, with hepatocytes and hepatic macrophages storing 10–20% of the total iron in the body. Spleen macrophages, bone marrow reticulocytes, and erythrocytes (the latter accounting for more than half of the total iron) account for the rest [21]. Hepatocytes’ iron storage capacity is due to their expression of a specific type of ferritin. Ferritins are composed of 24 subunits that can be either H- or L-type. The ratio of H- to L-type subunits determines the efficiency of iron storage/mobilization. Hepatic ferritins are composed of 80–95% L-type subunits, allowing for better iron storage than ferritins with more H-type subunits. For further details about ferritin structure and function, please see the review from Koorts and Viljoen [22].

The liver is also a key regulator of serum iron levels, through the secretion of the iron regulatory hormone hepcidin [23]. Secreted, hepcidin binds to FPN-1 at the basolateral membrane of enterocytes, driving its internalization and lysosomal degradation, thus inhibiting iron release [24]. Hepatic hepcidin expression is tightly regulated by serum iron levels. Increases in serum iron following intestinal uptake are sensed by hepatocytes via transferrin receptors 1 and 2 (TfR1/2). Although all iron-consuming cells express TfR1, allowing iron entry into the cell, only a few specialized cell types express TfR2, including hepatocytes.

At the surface, Fe^3+^-loaded Tf (holo-Tf) competes with the membrane-associated hereditary hemochromatosis protein (HFE) to bind both TfR1 and TfR2. When TfR1 is saturated with holo-Tf, TfR2 can then bind excessive holo-Tf along with HFE. This specific double interaction leads to TfR2-dependent hepcidin upregulation. In addition, in the hepatic HepG2 cell line, holo-Tf enhances the phosphorylation of ERK1/2 upon binding to the HFE–TfR2 complex, which in turn induces furin expression. Furin then enhances the maturation of both hepcidin and bone morphogenetic proteins (BMPs) by cleaving their nonmature forms, thus participating in a positive feedback loop, increasing hepcidin expression [25,26]. Therefore, whereas TfR1 delivers iron to every iron-consuming cell (including hepatocytes), the HFE–TfR2 complex appears to be a sensitive hepatic iron sensor.

Although it is the central mechanism for sensing serum iron levels, HFE–TfR2 interaction is not the only factor controlling hepatic hepcidin expression. Although these pathways have not been completely delineated, BMP6, interleukin 6 (IL-6), and the hypoxia-inducible factor 2α (HIF-2α) have all been implicated in the regulation of hepcidin expression.

Of the BMPs expressed in the liver, BMP6, whose expression depends on serum iron levels, is a potent inducer of hepcidin expression [27,28,29,30]. Upon the binding of BMP to a BMP receptor complex (BMPR1A or BMPR1B dimerized with BMPR-II), the receptor autophosphorylates and activates small mothers against decapentaplegic 1, 5, and 8 (SMAD1/5/8, the so-called R-SMADs). Phosphorylated R-SMADs bind to SMAD4 and translocate to the nucleus, where they activate the transcription of *HAMP*, the gene coding for hepcidin [27,31]. As a consequence, mice lacking *Bmp6* display severe iron overload [32]. BMP6 regulates hepatic hepcidin expression, since iron overload leads to increased *Bmp6* expression in the liver, but not in the duodenum (where iron absorption occurs) in mice [29]. BMP6-induced hepcidin expression is regulated by the hemojuvelin protein (HJV). *Hjv*^−/−^ mice develop severe iron overload due to the disruption of BMP6-dependent hepcidin expression [30]. HJV acts as a co-receptor for the BMP receptor complexes and is required for the full upregulation of hepcidin expression by BMP6 [28]. However, iron deficiency increases hepatic MTP-2 and furin expression through the stabilization of HIF-1α, which in turn cleave and inactivate membrane-anchored HJV to the soluble form of HJV (sHJV) [33,34]. sHJV is then released from the hepatocyte membrane and competes with the BMP receptor by sequestering BMPs, thus reducing hepatic hepcidin expression [35,36,37]. Interestingly, *Hfe*^−/−^ mice show decreased R-SMAD phosphorylation, which can be reversed by BMP6 overexpression, suggesting that HFE may be an enhancer rather than an activator of SMAD signaling [38,39,40].

Inflammation and its mediators are known to decrease serum iron levels, regardless of iron status. IL-6 plays a predominant role in this process through the upregulation of hepatic hepcidin expression, which may account for the hepcidin-driven iron deficiency observed in certain autoimmune and inflammatory diseases [41]. The binding of IL-6 to its membrane-anchored receptor (IL6R, or CD126) induces the assembly of an IL6R complex made of two IL-6-bound IL6Rs and a dimer of gp130 proteins [42]. This triggers the autophosphorylation of IL6R-associated Janus kinase 2 (JAK2), which then phosphorylates the signal transducer and activator of transcription 3 (STAT3). Activated STAT3 dimers translocate to the nucleus, where they bind to the promoters of target genes, including *HAMP*, allowing their transcription. Interestingly, IL-6-dependent hepcidin expression requires Alk3 and SMAD4, indicating that IL-6 may regulate hepcidin expression through SMAD signaling [27,31,43,44].

### 2.3. Kidney

The kidneys also play a role in iron homeostasis. Iron reaches the renal corpuscle by glomerular filtration and is reabsorbed in the renal tubule. During iron overload events, FPN-1 is expressed in the renal tubule to support the excretion of iron [45]. Renal cells also express hepcidin, and renal tubular epithelial cells are the main oxygen sensors in the kidneys [46]. In these cells, hypoxia-inducible factor (HIF) alpha subunits are hydroxylated by oxygen- and iron-dependent prolyl hydroxylases (which contain an iron atom within their catalytic site), allowing them to interact with the von Hippel–Lindau protein (pVHL), an E3 ubiquitin ligase. This complex is then ubiquitinated and labeled for proteasomal degradation. In contrast, during hypoxia, prolyl hydroxylases are inhibited, leading to HIF stabilization. In renal epithelial tubular cells, HIFs induce erythropoietin (EPO) production [23,47]. EPO stimulates erythropoiesis through the downregulation of hepcidin expression through an indirect mechanism involving erythroferrone (ERFE) production by the erythroblasts (Figure 1) [48,49]. In addition to their indirect effect on *HAMP* expression, HIFs regulate the expression of other iron-related genes, such as *CXCL12* (SDF-1α), *HGF* (hepatocyte growth factor), *CYBRD1* (Dcytb), *SLC11A2* (DMT-1), and *SLC40A1* (FPN-1) [50]. Thus, kidney-secreted EPO plays a non-negligible role in iron homeostasis, in particular through the decrease of hepatic hepcidin expression.

### 2.4. Scavenging Macrophages

Iron present in senescent and dying red blood cells can be recycled by spleen, liver, and bone macrophages. Typically, hemoglobin binds to its carrier protein haptoglobin, while free heme groups bind to hemopexin. These complexes bind in turn to CD163 or low-density lipoprotein receptor-related protein 1 (LRP1, also called CD91), respectively, at the surface of macrophages (Figure 1). They are then internalized for lysosomal degradation, where acidification by H^+^-ATPases oxidizes heme, releasing iron from the protoporphyrin ring [51]. Once in the cytosol, iron can either be stored bound to ferritin or exported through FPN-1 in response to decreases in hepcidin serum levels.

### 2.5. Cellular Iron Sensing, Consumption, and Functions

Every cell type needs iron, which is coordinated by Tf and TfR1. Once saturated, circulating holo-Tf binds to TfR1 (and TfR2 for a few cell types) at the cell surface, which mediates clathrin-dependent endocytosis of the complex. In endosomes, H^+^-ATPase-mediated acidification dissociates Fe^3+^ from Tf. Fe^3+^ is then reduced to Fe^2+^ by the STEAP (six-transmembrane epithelial antigen of the prostate) family of proteins (STEAP3 in particular) and then exported to the cytoplasm through DMT-1 to join the labile iron pool (LIP) [52] (Figure 1). Fe^2+^ can be stored bound to ferritin or utilized by the mitochondria or activate iron regulatory proteins (IRP-1/2). Mitochondrial uptake of iron is mediated by the iron transporters, mitoferrins (MFRN1/2). In the mitochondria, iron is metabolized by the iron–sulfur cluster (ISC) assembly machinery, which is responsible for the biogenesis of ISC. ISC-containing proteins (ISPs) are present in the mitochondria, the cytoplasm, and the nucleus, linking the mitochondria to cellular iron homeostasis (e.g., IRP-1 contains an ISC necessary for its function; see below). While cytoplasmic and nuclear ISPs are critical for vital cellular processes (i.e., DNA synthesis, repair and transcription, amino acid synthesis and protein translation, and vitamin and heme synthesis), mitochondrial ISPs are involved in the ISC assembly process itself, which requires a functional electron transfer chain including NADPH and complex I [6,8,50,53,54]; for further details on iron processing by the mitochondria, please see Lill et al. [54]. ISCs serve as cellular iron sensors establishing a feedback loop to avoid excessive iron uptake. At physiological ranges of cellular iron, ISCs bind to iron regulatory protein 1 (IRP-1), which hides the iron-responsive element (IRE) binding site of IRP-1 and gives it an aconitase function (i.e., the family of enzymes that catalyze the isomerization of citrate to cis-aconitate and then to D-isocitrate within the citric acid cycle). When cellular iron and ISC levels drop, IRP-1 binds to IRE sequences in target mRNAs [55]. At lower iron levels, the ubiquitin ligase FBXL5 also labels IRP-2 for proteasomal degradation [56]. Although IRP-1/2 are differentially regulated, they both promote their translation by binding to an IRE in the 3′ UTR region of target mRNAs. In contrast, the binding of IRP-1/2 to IREs in the 5′ UTRs aborts translation. More specifically, IRP-1/2 enhance TfR expression and inhibit ferritin, FPN-1, and HIF-2α expression [55,57]. Of note, iron transport is the same in endothelial cells (ECs), allowing the reciprocal passage of iron between tissues and the circulation [58].

## 3. Iron Status in PAH and Experimental PH

PAH is a rare cardiopulmonary disease associated with 30% mortality within 3 years of the diagnosis [59]. PAH is characterized by intense obliterative remodeling of pulmonary vasculature, increasing mean pulmonary artery pressure (≥20 mmHg) and pulmonary vascular resistance (>3 WO) (Figure 2). The raised afterload results in right ventricle (RV) hypertrophy, and ultimately cardiac dysfunction and failure, leading to death [60].

The prevalence of the disease is about 15–50 cases per million. Diagnosis is delayed by the nonspecific symptoms (mainly arising from right ventricular dysfunction): fatigue, dyspnea, chest pain, and syncopal episodes. PAH etiology is incompletely understood but includes idiopathic PAH (IPAH), heritable PAH (HPAH, due to mutations in BMPR2, BMPR1B, SMAD9, CAV1, KCNK3, and EIF2AK4), drug- or toxin-induced PAH (e.g., chemotherapeutic agents and occupational exposure to solvents), associated PAH (i.e., PAH associated with connective tissue disease, congenital heart disease, portal hypertension, HIV infection, and schistosomiasis), and pulmonary veno-occlusive disease [60]. The penetrance of PAH is incomplete; therefore, the pathogenesis might involve at least two hits (e.g., a predisposing genetic mutation associated with an environmental cause). In all cases, PAH involves pulmonary endothelial dysfunction leading to an imbalance between vasoconstrictor and vasodilator secretion, so the smooth muscle layer is overconstricted. Current therapeutic routes target this vasoconstriction by the endothelin (e.g., endothelin receptor antagonists), nitric oxide (e.g., exogenous nitric oxide, soluble guanylate cyclase stimulators, and phosphodiesterase 5 inhibitors), and prostacyclin (PgI2) (e.g., PgI2 analogues and nonprostanoid prostaglandin I receptor agonists) pathways [60]. Despite significant improvements in quality of life and survival, these medications do not reverse the structural obstructive remodeling (Figure 2). Thus, the only therapeutic option at late stages is lung transplantation.

### 3.1. Iron Levels and Manipulation in PAH

Decreased oxygen availability induces hypoxic pulmonary vasoconstriction along with secretion of EPO by the kidneys to increase erythropoiesis, an adaptation that tends to restore acceptable oxygen delivery to tissues when alveolar hypoxia occurs [61]. Moreover, the rise in pulmonary artery systolic pressure (PASP) observed during hypoxia is related to iron levels, since intravenous administration of the iron chelator deferoxamine (DFO) increased RVSP in both normoxic and hypoxic conditions in healthy individuals [62,63]; DFO also induces increased levels of EPO, mimicking the hypoxia response of the pulmonary vasculature [64]. Intravenous administration of iron to volunteers in hypoxia induced a 50% drop in PASP, which was still observed when they were exposed to a new hypoxic event 43 days later [65]. Thus, iron deficiency mimics hypoxia in normoxic conditions and worsens its adverse effects. Iron replacement reduces hypoxic pulmonary vasoconstriction and hence pulmonary hypertension. Although interesting, these studies focused on group 3 pulmonary hypertension, but the story complicates in the context of PAH (group 1). Serum iron levels are decreased in both IPAH and HPAH patients, which correlates with poor outcomes [10,11,12]. While hypoxic pulmonary vascular response mainly involves hepcidin downregulation due to HIF activation, in PAH, iron levels are low mainly because of hepcidin upregulation [10,11,63]. Thus, while it is tempting to restore iron levels in PAH patients, Ruiter and collaborators failed to increase patients’ serum iron levels (only 2 out of 18) by oral iron supplementation due to a hepcidin-related absorption defect [11]. Despite this expected pitfall, an open-label phase III clinical trial was launched (clinicaltrials.gov; identifier: NCT03371173; see Table 1) aiming to study the effects of oral iron supplementation in a larger cohort of PAH patients. However, this clinical trial was terminated due to low patient enrolment. In contrast, intravenous iron supplementation resulted in improved quality of life and increased skeletal muscle exercise capacity, which was attributable to better oxygen transport within the skeletal muscle, since myoglobin levels and mitochondrial oxygen capacity were enhanced in the quadriceps of iron-treated PAH patients [66,67]. Thus, while oral iron supplementation does not fix iron deficiency in PAH because hepcidin overexpression impairs iron absorption, intravenous iron administration improves some aspects of the disease. Recruitment of iron-deficient PAH patients for a 24-month multicenter, placebo-controlled, phase II clinical trial (clinicaltrials.gov; identifier: NCT01447628; see Table 1) to assess the effect of intravenous iron supplementation on cardiopulmonary hemodynamics, exercise capacity, and quality of life has recently been completed, which should help clarify the issue.

A critical point is that iron deficiency is measured by different methods in clinical trials (e.g., serum ferritin, transferrin, and soluble transferrin receptor (sTfR) dose; transferrin saturation; and iron itself, with or without second measures, such as vitamin D or C-reactive protein) that cannot be compared, as they are not directly correlated and may evolve independently of each other. Therefore, the lack of a standardized diagnostic method precludes a definite assessment of the impact of iron manipulation in PAH. According to a recent retrospective study, the ratio of $\frac{sTfR}{\log\left(serum\;Ferritin\right)}$ (the sTFRF index) corrected for CRP level might be a better prognostic biomarker to assess iron deficiency in PAH patients, and future studies may consider this issue. Iron deficiency is defined as a sTFRF index > 3.2 if CRP < 0.5 mg/dL or a sTFRF index > 2 if CRP > 0.5 mg/dL [68]. In addition, the three trials manipulating iron in PAH patients are open-labeled and enrolled only 15 to 21 iron-deficient PAH patients. Neither Inflammation status nor hepcidin levels were systematically studied in the enrolled patients. Further clinical trials enrolling more patients and studying all the confounding parameters would be beneficial to identify potential subgroups with different responses to iron therapy.

Despite their limitations and discrepancies, the three studies have some common findings. First, sex, age, menstruations, and anemia are not confounding effects on iron status in either the control subjects or the PAH patients [10,11]. Although a small subset of iron-deficient PAH patients have very low hepcidin levels, the vast majority of iron-deficient PAH patients show abnormally high hepcidin levels [10]. PAH patients also display raised sTfR levels, which may represent an adaptive mechanism to counteract increased hepcidin [10]. It is thus likely that iron deficiency arises from raised hepcidin levels. In addition, HIF stabilization and EPO expression are increased in PAH, which may indicate active erythropoiesis, which is inconsistent with raised hepcidin levels. Therefore, it might be interesting to study EPO and hepcidin expression in larger cohorts.

Finally, rare PH patients present mutations in genes involved in iron metabolism. For instance, a 29-years-old woman with homozygous ISCU (iron–sulfur cluster assembly enzyme of the mitochondria) mutations associated with hereditary myopathy with lactic acidosis and PH (mPAP = 21 mmHg at rest) showed exercise-induced pulmonary vascular dysfunction [69]. Mutations in BOLA3 (BolA family member 3), a gene coding for a protein essential for Fe–S maturation downstream of ISCU1/2, can lead to multiple mitochondrial dysfunction syndrome associated with PH [70]. Mutations in NFU1 (NFU1 iron–sulfur cluster scaffold homolog), a gene coding for a protein that delivers Fe–S clusters to proteins of the mitochondrial respiratory complexes I and II, are associated with PH [71]. Homozygous VHL mutations leading to congenital polycythemia are also associated with PAH [72,73,74,75]. Such cases may account for non-iron-deficient PAH patients and/or hepcidin-independent iron-deficient PAH patients. Therefore, they may explain, at least in part, different responses of intravenous iron supplementation in case of iron deficiency among the abovementioned clinical trials. Of note, an observational and prospective clinical trial to determine whether low iron–sulfur cluster levels, as it is the case in iron deficiency, can cause PH (clinicaltrials.gov; identifier: NCT02594917; see Table 1) is currently recruiting. Such an association would support iron deficiency, and Fe–S cluster deficiency in particular, as a cause/contributor of PAH development.

### 3.2. Iron Levels in Experimental PH

Early reports showed that unlike human PAH, continuous administration of monocrotaline (MCT), a pyrrolizidine alkaloid that induces pulmonary endothelial cell damage leading to pulmonary arterial hypertension, for 6 weeks in rats did not result in decreased serum iron levels despite severe PH and RV hypertrophy [76,77]. Later findings confirmed the lack of changes in serum iron levels, although both the pulmonary arteries and RV of rats with MCT-induced PAH showed increased TfR1 levels, which are normally associated with decreased iron levels [78]. In mice, lung ferritin and TfR1 levels were not changed in hypoxia-induced PH, while serum iron was not measured [79].

Adding to its complexity, serum iron and most iron signaling mediators are affected by sex in experimental animals and could depend on animal strain [80,81,82]. Overall, there is still a lack of clear experimental evidence on iron in rat and mouse PAH.

### 3.3. Iron Manipulation in Experimental PH

The potential contribution of iron deficiency to the pathogenesis of PH has been tested in preclinical models either supplementing or chelating iron to explore the role of its deficiency in disease development. DFO was shown to prevent the development of chronic hypoxia-induced PH in rats, while inhibiting the proliferation of cultured bovine pulmonary artery smooth muscle cells (PASMCs), suggesting a link between iron-induced oxidation and PH development [83]. However, the authors did not report on the iron status after 2 weeks. Using an alternative model, Naito et al found that a low iron diet prevented the development of MCT-induced PH, inhibiting both pulmonary and right ventricular remodeling and increasing survival. However, as previously mentioned, MCT animals did not show altered serum iron concentration and, therefore, did not recapitulate the low iron status of PAH [10,11,78]. This could be due to the faster disease development in animal models not allowing time for the establishment of iron deficiency. These findings suggest that decreasing iron levels may be beneficial in experimental PH; however, since the animal models used do not replicate the iron deficiency observed in patients with PAH, the therapeutic potential of restoring iron homeostasis is unclear.

Since patients with PAH have an iron deficiency, an attempt was made to determine its pathophysiological role in the development of the disease [82]. A work performed in rats showed that an iron-deficient diet resulted in the development of PH and pulmonary vascular remodeling remarkably similar to the human disease. Chelation of iron in vitro resulted in increased human PASMC proliferation, which was restored with iron supplementation [84]. Ultimately, intravenous iron supplementation attenuated PH development in iron-deficient rats, demonstrating the potential role of iron therapy in PAH with iron deficiency [84]. Unlike previous reports, this work showed that iron supplementation is beneficial when iron deficiency is established. A recent study conducted in mice showed that iron deficiency specifically in PASMCs (by disruption of the hepcidin binding site on FPN) is also implicated in the development of experimental PH by inducing proliferation and ET-1 expression in these cells [85]. Importantly, this study also showed that specific PASMC iron deficiency is sufficient to develop PH in these mice, suggesting that both systemic and local iron deficiency have a non-negligible role in the PAH pathogenesis [85]. It has long been shown in PAH that pulmonary vascular cells, in both patients and animal models, present mitochondrial abnormalities, leading to a shift of these cells from oxidative respiration to glycolysis, a mechanism called the Warburg effect [86,87]. In accordance with these metabolic changes, previous reports attest that hypoxia-induced miR210 expression leads to ISC deficiency and decreased mitochondrial respiration in murine and human pulmonary vascular cells [88]. Further studies in the rat showed that genetic disruption of ISCU1/2 or BOLA3 mimics hypoxic conditions (which mediates HIF-dependent ISCU1/2 and BOLA3 downregulation) and eventually leads to PH [69,70,88]. These rat models link further between the observed Warburg effect in the pulmonary vascular cells of PAH patients and iron deficiency [87]. Such studies may also explain the association between some metabolic diseases (due to ISCU1/2 or BOLA3 mutations in particular) and PH [87].

Finally, it is not likely that iron deficiency alone would explain pulmonary hypertension development in patients, and these particular models may rather mimic hypoxic conditions, which induce both experimental and human PH. Further studies are needed to better understand these results.

### 3.4. Role of Iron in the RV

After 4 days of hypoxia (10% O_2_), iron chelation in rats decreased RV hypertrophy and GATA4 expression, which plays a major role in its adaptation to pressure overload [89,90]. In fact, iron deficiency promoted skeletal muscle atrophy, while iron supplementation had the opposite effect, suggesting that iron is required for muscle adaptation to overload [91]. Myoglobin, which serves as a skeletal muscle iron store, plays an important role in the adaptation of the RV to pressure overload [92]. Although no studies have directly examined the role of iron deficiency on the RV, a previous report found decreased RV myoglobin content, which indicates cardiac iron deficiency, since iron is required for myoglobin production, which was associated with reduced RV capillarization in both patients with PAH and rats with stable and progressive MCT-induced PH [93,94]. Indeed, in rodents, iron deficiency compromises exercise capacity and is associated with skeletal muscle alterations, including decreased myoglobin, which are corrected by iron supplementation [95]. In vitro iron deficiency also leads to compromised contractility in human stem cell-derived cardiomyocytes, which was associated with decreased mitochondrial function, and could be reversed by iron supplementation [96]. The translation of these findings using IPSC-derived cardiomyocytes from patients with PAH would be invaluable [97]. Furthermore, the connection between BMP signaling and hepcidin expression, along with the recent observation that rats with a *BMPR2* mutation show intrinsic RV cardiomyocyte dysfunction, could explain the lower cardiac index and increased mortality in patients with iron deficiency [27,28,68,98].

The limited evidence suggests a beneficial role for iron supplementation in preclinical PH with iron deficiency. Nonetheless, while treating rats with a nontoxic and moderate iron dose was not associated with any cardiopulmonary changes, severe chronic iron overload (200 mg/kg iron dextran for 28 days) resulted in RV hypertrophy and increased pulmonary vascular reactivity and remodeling [99]. These changes did not parallel increased RV pressure and were associated with mild right heart systolic and diastolic dysfunction at the highest iron regimen. The deleterious effects of iron overload could hinder the use of iron supplementation in PAH, but this can be balanced by meticulous hemodynamic assessment and alternative formulations and targets [100,101,102]. As the word suggests, overload does not equal homeostasis, and both overload and deficiency might show similar deleterious effects, suggesting that correcting iron levels in iron-deficient patients with PAH would require careful monitoring [103].

## 4. Molecular Links between Iron Deficiency and PAH

Approximately 40% of PAH patients have iron deficiency due to hepcidin-dependent iron malabsorption [10,11,12]. Interestingly, it has recently been shown that the hepcidin–ferroportin axis also enhances the proliferation of pulmonary artery smooth muscle cells (PASMCs), suggesting the direct participation of hepcidin in pulmonary vascular remodeling in PAH (Figure 2) [85,104]. PAH is a multifactorial disease, and inflammation, endothelial dysfunction, hypoxia, and vasoconstriction have all been implicated in its development. In this section, we explore whether iron deficiency is linked to PAH pathophysiology at the molecular scale.

### 4.1. Inflammation

Inflammation is a prominent feature of PAH [105]. Innate and adaptive immune cells accumulate in the vicinity of remodeled pulmonary arteries, where they produce and release inflammatory mediators (for further details, see [106]). All types of PAH vascular lesions are associated with inflammatory cell infiltrates [107]. Beyond inflammation, the connection of remodeled vessels to tertiary lymphoid tissues (tLTs), the presence of immunoglobulin deposits in the lung, and circulating autoantibodies directed to vascular wall components all suggest the involvement of adaptive and autoimmune responses [105,108]. Inflammation and autoimmunity in PAH may contribute to iron deficiency. Mammals have developed multiple innate immunity mechanisms that limit the availability of iron to pathogens upon infection. Thus, anemia has been established as an effect of host defense in inflammation and chronic diseases [109]. Mechanistically, inflammation has been shown to upregulate hepcidin expression.

IL-6 and IL-1β are major mediators of inflammation in PAH, which are abnormally high in patients. Serum IL-6 levels also correlate with the severity and prognosis of the disease [110,111]. Increased levels of circulating IL-6 may activate sIL6R and membrane-bound IL6R in hepatocytes, increasing hepatic IL-6 signaling, leading to STAT3 activation and IL-6-dependent BMP signaling, further increasing hepcidin levels (Figure 3). However, studies by Soon et al. [12] and Rhodes et al. [10] failed to find a correlation between IL-6 and hepcidin levels, although iron and IL-6 levels were correlated, suggesting that IL-6 may not be the sole contributor to raised hepcidin levels in PAH. In particular, IL-1β is also known to increase hepatocyte hepcidin expression, especially in the absence of IL-6 signaling, so it might contribute to the increase in circulating hepcidin levels in PAH [112].

### 4.2. BMPR-II Signaling

Monoallelic loss-of-function mutations in the gene encoding bone morphogenetic protein receptor 2 (*BMPR2*) are the main genetic risk factor for heritable pulmonary arterial hypertension [113]. Moreover, the expression of BMPR-II is decreased in idiopathic PAH [114]. BMPR-II dysfunction favors TGF-β pathway activation and subsequently increases PASMC proliferation and apoptosis resistance (Figure 4).

In the hepatic HepG2 cell line, knockdown of BMPR-II was found to increase hepcidin expression, raising the possibility that *BMPR2* mutations in patients with PAH account for the increased serum hepcidin levels [10]. Paradoxically, mice with hepatic disruption of *BMPR2* or *ACVR2A* (ActRIIa, another BMP type 2 receptor) display normal iron homeostasis and hepcidin expression, whereas mice lacking both *BMPR2* and *ACVR2A* have strong iron overload and very low hepcidin expression. These surprising findings suggest redundant control of hepatic hepcidin expression by Alk1/2/3/BMPR-II and Alk1/2/3/ActRIIa [44]. BMP6 and BMP7 also appear to play redundant roles in stimulating hepcidin expression. Certain BMPR-II mutations induce a BMP7 gain of function, increasing hepatic hepcidin expression in mice, and administration of exogenous BMP7 corrects iron overload in *Bmp6*-deficient mice [115,116,117]. Interestingly, BMP7 is overexpressed in PAH and correlates with a high mortality risk [118]. Therefore, it is likely that BMP6 and BMP7 upregulate hepcidin expression upon binding to Alk1/2/3/BMPR2 or Alk1/2/3/ActRIIa, suggesting that another type 2 BMP receptor (i.e., ActRIIa) may take over for BMPR-II in the case of a BMPR-II loss-of-function mutation [44] (Figure 3 and Figure 4).

### 4.3. Hypoxia-Inducible Factors

PAH patients have raised circulating and pulmonary levels of EPO [119]. As previously discussed, iron deficiency mimics tissue hypoxia and thus leads to the stabilization of HIFs. In renal tubular epithelial cells, HIF-2α induces EPO secretion, which in turn stimulates the proliferation of both pulmonary vascular endothelial and smooth muscle cells, contributing to the pathogenesis of the disease [119]. In addition, despite a lack of clinical reports, PAH patients are likely to suffer from hypoxemia, at least during sleep apnea, which would cause tissue hypoxia [120,121]. Decreased cardiac output and the subsequent weak perfusion of the systemic organs may also contribute to tissue hypoxia [122]. Therefore, regardless of the etiology of the disease, PAH might involve an intermittent systemic hypoxia and/or an iron deficiency mimicking hypoxia, which would increase HIF stabilization in the pulmonary vasculature and the kidneys [122,123]. In the kidneys, HIF-2α induces the secretion of EPO and thus stimulates erythropoiesis (Figure 1). However, in PAH, erythropoiesis is limited due to the elevated hepcidin levels, despite the concomitant rise in EPO levels [10,11,119]. This may lead to enhanced but inefficient erythropoiesis, as attested by the iron deficiency anemia and erythrocytosis in both patients and mice [60,75,124,125]. Indeed, PAH patients have an increased CD34^+^ CD133^+^ erythroid progenitor population compared with healthy subjects [123]. In addition, deletion of IRP-1 induces pulmonary hypertension (PH) and polycythemia in mice through the stabilization of HIF-2α, which increases ET-1 expression in pulmonary endothelial cells, further contributing to disease progression [126]. A recent study developed a new model of severe experimental pulmonary hypertension that recapitulates pulmonary vascular remodeling, occlusion, cardiac remodeling, and heart failure in mice by disrupting HIF prolyl hydroxylases, which stabilizes HIF-2α specifically in pulmonary endothelial cells and hematopoietic cells [127]. Thus, hypoxia triggers both iron-consuming erythropoiesis and pulmonary vascular cell proliferation through the stabilization of HIF-2α (Figure 1 and Figure 2).

In addition, the circulating level of macrophage inhibitory cytokine 1 (MIC-1, corresponding to the murine GDF-15), a member of the TGF-β superfamily, is elevated in the serum of PAH patients [128,129]. Although MIC-1 may directly contribute to PAH through its TGF-β-like properties, its expression increases critically in iron-deficient patients, regardless of PAH, and in iron-depleted healthy volunteers [130]. Red cell distribution width (RDW) is a routine blood test that estimates the variation in red blood cell volume and size. MIC-1 levels and RDW are increased in IPAH and are potent biomarkers for predicting patient survival, further indicating that iron-deficient erythrocytosis may occur in PAH and contribute to disease progression [128,129,131]. The fact that PAH patients show increased circulating endothelial progenitor cells is consistent with increased RDW [123]. In contrast, the increased MIC-1 and EPO levels, both of which downregulate hepcidin expression, cannot explain the high levels of hepcidin in PAH patients and probably reflect adverse cardiac events and hypoxic and/or iron-deficient conditions. This apparent disconnection between EPO and hepcidin levels has also been highlighted in response to different hypoxic conditions, and high-altitude adaptation in particular [132]. In both good and bad adaption to altitude, EPO levels are high, whereas hepcidin levels are low, as expected, in case of good adaptation but high in case of bad adaption. Even if underlying mechanisms of such loss of EPO-mediated hepcidin inhibition are still not known, it confirms that uncontrolled hepcidin expression is linked to pathologic states.

In summary, hepcidin-dependent iron deficiency in PAH is mainly due to impaired BMP signaling and the proinflammatory IL-6 axis (Figure 3).

## 5. Clinical Implications

Iron deficiency has been shown to correlate with poor survival in PAH [10,11,68]. As discussed above, intravenous administration of iron to PAH patients increases exercise endurance capacity, while oral supplementation has little effect due to low intestinal iron absorption in the presence of high levels of hepcidin [13]. In heart failure characterized by iron maldistribution, intravenous administration of iron, even in the absence of anemia, improves physical performance, New York Heart Association (NYHA) functional class, and quality of life compared with a placebo [133]. Thus, iron may have protective effects on the heart itself. The hypothesis that intravenous iron may correct cardiomyocyte iron deficiency in the RV maladaptation seen in PAH is intriguing and needs to be further explored. However, iron supplementation could have adverse effects and should be considered carefully. Indeed, most bacteria require iron for growth and survival. Multiple studies have shown that iron supplementation alters the gut’s microbial profile, promoting the growth of pathogenic enterobacteria species at the expense of protective lactobacilli and bifidobacteria species [134]. Thus, iron supplementation needs to take into account potential effects on the microbiome. In particular, it could aggravate inflammation and favor opportunistic infections, which may worsen the disease. Hepcidin itself may directly increase PASMC proliferation, thus participating in pulmonary vascular remodeling and disease progression [104]. Therefore, modulation of the hepcidin/ferroportin axis could provide a more powerful and safer alternative to iron supplementation. There are several existing and emerging treatment options that seek either to stabilize ferroportin expression and activity in spite of excess hepcidin or to target hepcidin directly using antagonists or neutralizing antibodies [135]. LY2787106, a monoclonal antihepcidin antibody developed by Eli Lilly and Company, was well tolerated and resulted in transient iron mobilization and increased reticulocyte count relative to baseline in a phase 1 clinical trial in patients with cancer-associated anemia (clinicaltrials.gov; identifier: NCT01340976; see Table 1) [136]. A comprehensive list of hepcidin antagonists that directly target hepcidin or the abovementioned pathways and have been validated in preclinical or clinical settings other than PAH is described and discussed in Sebastiani et al. [135]. Other possible strategies focus on silencing the IL-6 pathway (e.g., blocking IL-6 or IL6R with monoclonal antibodies or inhibiting STAT3 or JAK2 with small molecules) [135]. Tocilizumab, an anti-IL6R monoclonal antibody, is currently under consideration in an open-label phase 2 clinical trial for PAH (clinicaltrials.gov; identifier: NCT02676947; see Table 1) [137]. Unfortunately, the investigators did not include measurements of iron levels in their trial plan; nevertheless, this treatment has been shown to reduce hepcidin levels and improve anemia in rheumatoid arthritis patients [138]. Finally, sotatercept, an ActRIIa ligand trap, has been designed to rebalance the BMP-/TGF-β-dependent signalings (Figure 4). In preclinical models, sotatercept inhibited SMAD2/3 activation in both lungs and RV, reversed pulmonary artery remodeling, and restored RV function [139]. A placebo-controlled phase II clinical trial has been designed to assess its efficacy and safety in adult PAH (clinicaltrials.gov; identifier: NCT03496207; see Table 1). This approach may correct SMAD-dependent hepcidin overexpression in PAH. Such approaches alone or in combination with iron supplementation may be beneficial in PAH and to counteract medial hyperplasia and remodeling of the pulmonary vasculature.

## 6. Summary and Conclusions

Iron is required for vital functions, including oxygen transport, DNA synthesis, and cellular metabolism. Accordingly, complex machinery has evolved to sense and regulate iron levels within the blood and the different tissues. The liver is a key regulator of iron levels in the blood through the secretion of the iron regulatory hormone hepcidin. Once secreted, hepcidin binds to FPN-1, the only known iron exporter, at the basolateral membrane of enterocytes, driving its degradation and inhibiting intestinal iron absorption and release into the circulation. Iron deficiency is observed in parasitic infections, cancers, anemia, neurodegenerative diseases, some genetic disorders, and cardiovascular diseases, such as PAH. PAH is a rare and fatal incurable disease characterized by increased pulmonary artery pressure due to vasoconstriction and progressive obstructive remodeling of the small pulmonary arteries, leading to right heart hypertrophy and ultimately cardiac failure and death. Approximately 40% of PAH patients have an iron deficiency mainly due to high hepcidin levels. Iron deficiency is correlated with poor survival, while sex, age, and anemia are not correlated with iron deficiency in PAH. However, whether iron deficiency contributes to or is a mere consequence of the disease remains debated. Despite being considered the central mechanism sensing serum iron levels, the TfR2–HFE-mediated pathway is not the only axis controlling hepatic hepcidin expression and, therefore, systemic iron homeostasis. BMPs (BMP6 and BMP7), inflammation including increased levels of IL-6, and HIF-2α are also implicated in the regulation of hepcidin expression. Since they are also major pillars of PAH pathogenesis, iron deficiency is likely a consequence of the disease. However, iron deficiency promotes pulmonary vasoconstriction, which is inhibited by iron supplementation, and hepcidin/ferroportin also enhances the proliferation of pulmonary artery smooth muscle cells. This suggests that iron deficiency and hepcidin participate in the vasoconstriction and pulmonary vascular remodeling responsible for PAH. Moreover, current animal models of severe PH do not show iron deficiency, indicating that iron deficiency is not the direct effect of high pulmonary artery pressure or RV failure, at least in the time frame of these models. Iron deficiency is crucial for disease progression and worsening, and several completed or ongoing clinical trials have been launched to study the impact of iron supplementation on PAH (clinicaltrials.gov; identifier: NCT01446848, NCT00952302, NCT03371173, NCT01288651, NCT01847352, NCT01447628; see Table 1). Completed studies indicate that oral iron supplementation does not fix iron deficiency in PAH because of hepcidin-mediated dietary iron malabsorption, but intravenous iron administration improves some aspects of the disease. It also appears that iron deficiency can be diagnosed by various incomparable methods, which preclude a definite assessment of the impact of iron manipulation in PAH.

In conclusion, treating iron deficiency by targeting hepcidin or IL-6 signaling or rebalancing the BMP-/TGF-β-dependent pathways represents innovative strategies that need to be addressed as complements to conventional therapies in robust clinical trials. 

On a cautionary note, iron supplementation is known to fuel inflammation and infection and could therefore aggravate pulmonary vascular inflammation and favor opportunistic infections. Finally, iron deficiency compromises RV adaptation to pressure overload in rodent models. This needs to be confirmed in humans by assessing right heart function and adaptation to PAH with iron deficiency or supplementation.

## Figures and Tables

**Figure 1 cells-10-00477-f001:**
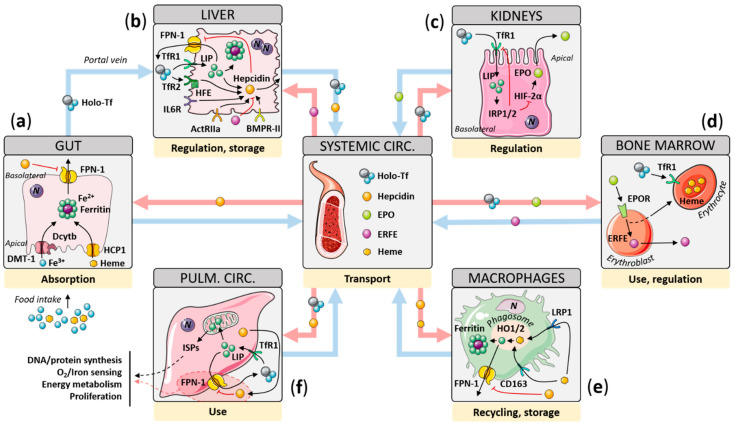
Schematic representation of systemic iron homeostasis. (**a**) Diet-derived iron is exported from enterocytes to the circulation through FPN-1, where it binds to carrier Tf. (**b**) When iron-charged Tf (holo-Tf) levels increase, holo-Tf binds to HFE/TfR2 complexes at the hepatocyte surface, which transduce a signal, leading to hepcidin expression and secretion. BMP, TGF-β, and IL-6 receptor activation also leads to hepcidin expression. The iron regulatory hormone hepcidin binds to and inhibits FPN-1, thus shutting down iron uptake and iron store mobilization. (**c**) In hypoxic and/or iron-deficient contexts, HIF-2α stabilizes in renal cells and increases EPO synthesis to increase erythropoiesis. (**d**) EPO promotes erythroblast maturation within the bone marrow, which leads to ERFE secretion. ERFE then negatively regulates hepatic hepcidin expression, thus enhancing iron-consuming erythropoiesis. (**e**) Hemes from senescent erythrocytes are captured by hepatic and splenic macrophages through LRP1 and CD163, and heme-derived iron is stored and/or exported to the circulation. (**f**) Holo-Tf binds to TfR1 in target cells, allowing its internalization and further iron dissociation. Besides, hepcidin enhances PASMC proliferation through the inhibition of FPN-1. ActRIIa: activin type IIa receptor, Alk2/3: activin receptor-like kinase 2/3, BMPR-II: bone morphogenetic protein receptor type II, EPO: erythropoietin, EPOR: erythropoietin receptor, ERFE: erythroferrone, FPN-1: ferroportin 1, HIF-2α: hypoxia-inducible factor 2α, HO1/2: heme oxygenase 1/2, holo-Tf: holo-transferrin, IL6R: interleukin-6 receptor, IRP1/2: iron regulatory protein 1/2, ISPs: iron–sulfur proteins, LIP: labile iron pool, LRP-1: low-density lipoprotein receptor-related protein 1, Tf: transferrin, TfR1/2: transferrin receptor 1/2.

**Figure 2 cells-10-00477-f002:**
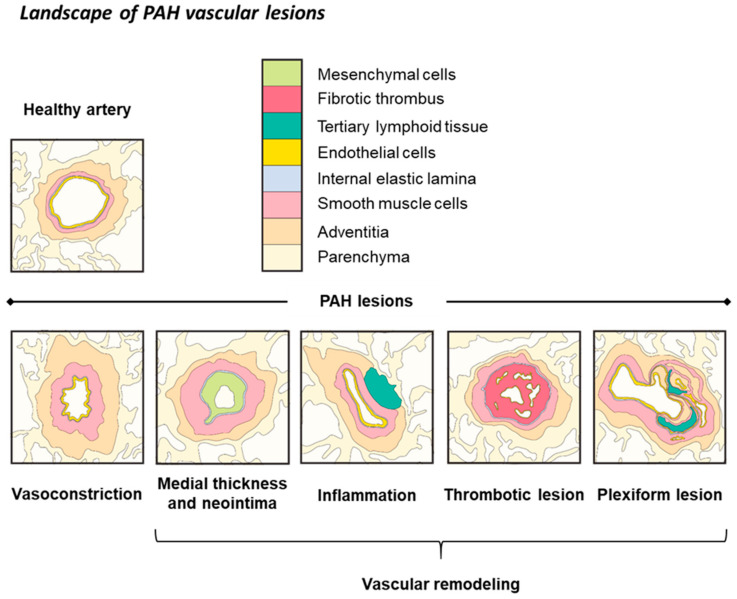
PAH pathophysiology. Healthy artery. Pulmonary circulation is a low-pressure, high-flow circuit that allows for rapid adaptation to effort. Pulmonary arteries have high compliance due to the thin media and low vascular resistance. PAH pathophysiology. PAH is a rare and fatal disease characterized by increased mean pulmonary artery pressure (≥20 mmHg) due to progressive obstructive remodeling of the small pulmonary arteries, leading to right heart hypertrophy and ultimately cardiac failure and death. The prevalence of the disease is about 15–50 cases per million. Diagnosis is delayed by the nonspecific symptoms (mainly arising from right ventricular dysfunction): fatigue, dyspnea, chest pain, and syncopal episodes. PAH etiology is incompletely understood but includes idiopathic PAH (IPAH), heritable PAH (HPAH, due to mutations in BMPR2, BMPR1B, SMAD9, CAV1, KCNK3, and EIF2AK4), drug- or toxin-induced PAH (e.g., chemotherapeutic agents and occupational exposure to solvents), associated PAH (i.e., PAH associated with connective tissue disease, congenital heart disease, portal hypertension, HIV infection, and schistosomiasis), and pulmonary veno-occlusive disease. The penetrance of PAH is incomplete; therefore, the pathogenesis might involve at least two hits (e.g., a predisposing genetic mutation associated with an environmental cause). In all cases, PAH involves pulmonary endothelial dysfunction leading to an imbalance between vasoconstrictor and vasodilator secretion, so the smooth muscle layer is overconstricted. Current therapeutic routes target this vasoconstriction by the endothelin (e.g., endothelin receptor antagonists), nitric oxide (e.g., exogenous nitric oxide, soluble guanylate cyclase stimulators, and phosphodiesterase 5 inhibitors), and prostacyclin (PgI2) (e.g., PgI2 analogues and nonprostanoid prostaglandin I receptor agonists) pathways. Despite significant improvements in quality of life and survival, these medications do not reverse the structural obstructive remodeling (see “Landscape of PAH vascular lesions”). Thus, the only therapeutic option at late stages is lung transplantation. PAH lesions. Pulmonary arteries of PAH patients are prone to vasoconstriction, mainly because of endothelial dysfunction, inflammation, and functional alterations of ions channels in PASMC. Impaired endothelial function, in particular, is responsible for the overproduction of vasoconstrictors (ET-1) and downregulation of vasodilators (prostacyclin, nitric oxide). This excessive vasoconstriction reduces the arterial lumen while increasing vascular resistance (>3 WO), leading to increased mean pulmonary artery pressure. Endothelial secretion of growth factors and proinflammatory cytokines, such as PDGF, EGF, IL-1β, and IL-6, as well as enhanced shear stress, drives important vascular remodeling: adventitial and medial thickening, neointima formation, and adventitial infiltration of immune cells. Endothelial injury also leads to local thrombosis and the formation of thrombotic lesion, which can be partially vascularized. Finally, the local proangiogenic microenvironment leads to aberrant intraluminal angiogenesis, which is responsible for the typical plexiform lesions.

**Figure 3 cells-10-00477-f003:**
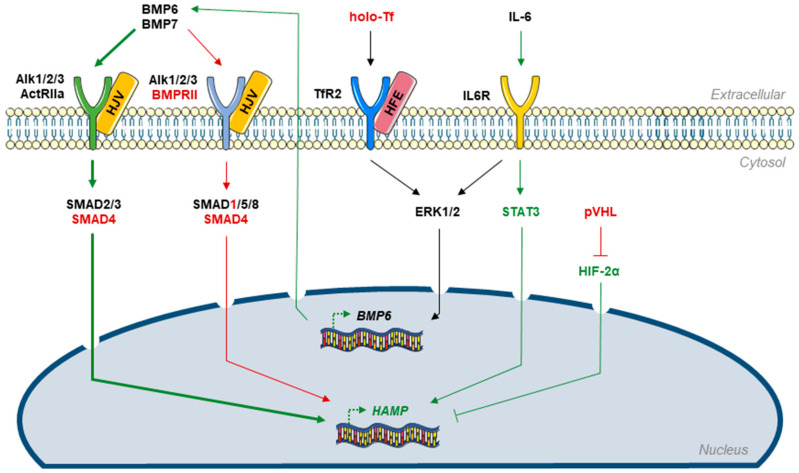
Dysregulated pathways governing liver hepcidin expression in PAH. When intestinal iron intake increases, holo-Tf level increases accordingly. High levels of holo-Tf are sensed by hepatocytes through HFE/TfR2 signaling, which induces BMP6 expression through ERK1/2 activation. Once secreted, BMP6 binds to TGF-β and BMP receptors, leading to SMAD signaling and hepcidin (HAMP) genetic expression. The IL6R pathway also induces ERK1/2 activation and further BMP6 expression in hepatocytes but can also activate HAMP transcription directly through STAT3 activation. In contrast, stabilized HIF-2α downregulates HAMP transcription and thus lowers hepatic hepcidin expression. Green arrows: expression or signaling enhanced in PAH; red arrows: expression or signaling decreased in PAH; green bold arrows: BMP signaling switch in PAH. ActRIIa: activin type IIa receptor, Alk2/3: activin receptor-like kinase 2/3, BMP: bone morphogenetic protein, BMPR-II: bone morphogenetic protein receptor type II, ERK1/2: extracellular signal-regulated kinase 1/2, *HAMP*: hepcidin-coding gene, HIF-2α: hypoxia-inducible factor 2α, holo-Tf: holo-transferrin, IL6R: interleukin-6 receptor, SMAD: small mothers against decapentaplegic, STAT3: signal transducer and activator of transcription 3, TfR2: transferrin receptor 2, pVHL: von Hippel–Lindau protein.

**Figure 4 cells-10-00477-f004:**
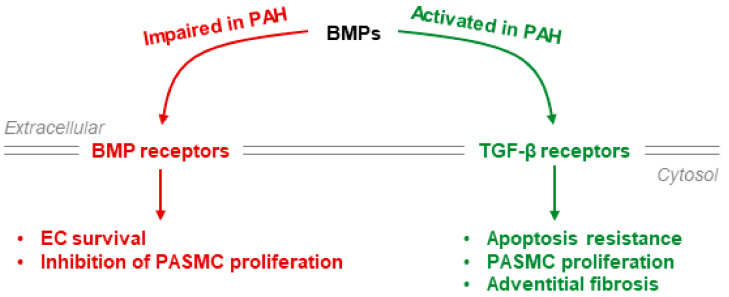
Imbalance between BMP and TGF-β signaling in PAH. BMPs can bind both BMP and TGF-β receptors. BMP receptor signaling leads to pulmonary endothelial cell survival and inhibition of PASMC proliferation, whereas TGF-β receptor activation promotes PASMC proliferation and survival and adventitial fibrosis.

**Table 1 cells-10-00477-t001:** Ongoing and completed clinical trials related to PAH and iron. The following queries have been submitted to https://clinicaltrials.gov/ (accessed on 18 February 2021): pulmonary arterial hypertension + iron; pulmonary arterial hypertension + iron deficiency; pulmonary arterial hypertension + iron supplementation; pulmonary arterial hypertension + hepcidin.

Clinical Trials	Status	ID
BMPR2 Mutations and Iron Metabolism in Pulmonary Arterial Hypertension	Recruiting	NCT04086537
IV Iron Replacement for Iron Deficiency in Idiopathic Pulmonary Arterial Hypertension (IPAH) Patients	Completed	NCT01447628
Iron Deficiency in Pulmonary Hypertension	Unknown status	NCT01288651
Oral Iron Supplementation in Pulmonary Hypertension	Completed	NCT01446848
Iron Status and Hypoxic Pulmonary Vascular Responses	Completed	NCT01847352
Eisenmenger Quality Enhancement Research Initiative	Completed	NCT01623492
Study of the Effects of Iron on Lung Blood Pressure at High Altitude	Withdrawn	NCT00960921
ORal IrON Supplementation with Ferric Maltol in Patients With Pulmonary Hypertension (ORION-PH-1)	Terminated	NCT03371173
Study of the Effects of Iron Levels on the Lungs at High Altitude	Completed	NCT00952302
Genetic and Environmental Determinants That Control Metabolism in Pulmonary Hypertension	Recruiting	NCT02594917
Blood Markers Predict Effect of Normobaric Hypoxia at Rest and during Exercise in Patients with Pulmonary Hypertension	Not yet recruiting	NCT04715113
A Study of Sotatercept for the Treatment of Pulmonary Arterial Hypertension (PAH)	Active, not recruiting	NCT03496207
Bardoxolone Methyl in Patients with Connective Tissue Disease-Associated Pulmonary Arterial Hypertension-CATALYST	Terminated	NCT02657356
Outcome Study Assessing a 75 Milligrams (mg) Dose of Macitentan in Patients with Pulmonary Arterial Hypertension	Recruiting	NCT04273945
Clinical Study of Pulsed, Inhaled Nitric Oxide versus Placebo in Symptomatic Subjects with PAH	Terminated	NCT02725372
Use of Inhaled Nitric Oxide to Prevent Pulmonary Hypertension Associated to Stored Blood Transfusion	Withdrawn	NCT02217683
Erythrocyte Glutamine Level Relation to Pulmonary Hypertension Risk in Beta Thalassemia Major Children	Completed	NCT03133169
Extended Access Program to Assess Long-Term Safety of Bardoxolone Methyl in Patients with Pulmonary Hypertension RANGER	Terminated	NCT03068130
Hydroxyurea and Erythropoietin to Treat Sickle Cell Anemia	Completed	NCT00270478
Beta3 Agonist Treatment in Chronic Pulmonary Hypertension Secondary to Heart Failure	Unknown status	NCT02775539
Effects of Inspiratory Muscle Training in Patients with Pulmonary Hypertension	Recruiting	NCT04152187
Sildenafil for Secondary Pulmonary Hypertension Due to Valvular Disease	Completed	NCT00862043
Efficacy, Safety, and Tolerability Study of Pirfenidone in Combination with Sildenafil in Participants with Advanced Idiopathic Pulmonary Fibrosis (IPF) and Intermediate or High Probability of Group 3 Pulmonary Hypertension	Completed	NCT02951429
AZ, MZ, and the Pulmonary System Response to Hypoxia	Completed	NCT02760121
Treatment of Atrial Fibrillation in Patients by Pulmonary Vein Isolation in Combination with Renal Denervation or Pulmonary Vein Isolation Only	Recruiting	NCT02115100
Hemodynamic Effects of Acute Normobaric Hypoxia during Exercise in Patients with Pulmonary Hypertension: Single-Center Randomized Controlled Trial	Not yet recruiting	NCT04715113
A Therapeutic Open Label Study of Tocilizumab in the Treatment of Pulmonary Arterial Hypertension (TRANSFORM-UK)	Completed	NCT02676947
A Phase 1 Study of LY2787106 in Cancer and Anemia	Terminated	NCT01340976

## Data Availability

No new data were created or analyzed in this study. Data sharing is not applicable to this article.

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
