# Peer review of "Iron Deficiency in Pulmonary Arterial Hypertension: A Deep Dive into the Mechanisms"

_cells, 2021, doi:10.3390/cells10020477_

Round 1

Reviewer 1 Report

The authors have compiled an extensive review on the relationship between iron deficiency and pulmonary arterial hypertension. Although the mechanism between the two has been reviewed in other journals relatively recently*, the topic is complex and this review offers new insights into both the molecular/cellular mechanisms and the current clinical interventions (*Sonnweber, T., Pizzini, A., Tancevski, I. et al. Anaemia, iron homeostasis and pulmonary hypertension: a review. Intern Emerg Med 15, 573–585 (2020), Ramakrishnan, Latha et al. “Pulmonary Arterial Hypertension: Iron Matters.” Frontiers in physiology vol. 9 641. 31 May. (2018)).  

I found that the overall organization and language of  the article was easy to follow despite the amount of information provided and that the figures offer a helpful overview of the topics addresses in the text.

I have very few minor comments:

Line 14: add affiliation for #5

line 60: consider changing the word machinery 

lines 96,97,98: consider rephrasing this very long sentence to increase clarity

lines 180, 181, 182: consider rephrasing to make clear that CD163 binds hemoglobin/haptoglobin while CD91 binds heme/hemopexin

Line 472: Consider rephrasing PAH "twisted" pathways

Author Response

The authors have compiled an extensive review on the relationship between iron deficiency and pulmonary arterial hypertension. Although the mechanism between the two has been reviewed in other journals relatively recently*, the topic is complex and this review offers new insights into both the molecular/cellular mechanisms and the current clinical interventions (*Sonnweber, T., Pizzini, A., Tancevski, I. et al. Anaemia, iron homeostasis and pulmonary hypertension: a review. Intern Emerg Med 15, 573–585 (2020), Ramakrishnan, Latha et al. “Pulmonary Arterial Hypertension: Iron Matters.” Frontiers in physiology vol. 9 641. 31 May. (2018)). 

I found that the overall organization and language of the article was easy to follow despite the amount of information provided and that the figures offer a helpful overview of the topics addresses in the text.

We thank reviewer 1 for his positive remarks.

Q1. Line 14: add affiliation for #5

R1. Affiliation 5 has been removed in the revised manuscript.

Q2. Line 60: consider changing the word machinery

R2. We changed the machinery-containing sentence by a less heavy one: “Since both iron overload and deficiency are deleterious, a tight regulation of iron balance is essential– from uptake in the gut to storage in the blood and liver (Figure 1)”

Q3. Lines 96,97,98: consider rephrasing this very long sentence to increase clarity

R3. The long sentence has been cut into two shorter ones.

Q4. Lines 180, 181, 182: consider rephrasing to make clear that CD163 binds hemoglobin/haptoglobin while CD91 binds heme/hemopexin

R4. We clarified these sentences in the revised manuscript: “Typically, hemoglobin binds to its carrier protein haptoglobin while free heme groups bind to hemopexin. These complexes bind in turn to CD163 or low density lipoprotein receptor-related protein 1 (LRP1 also called CD91) respectively, at the surface of macrophages (Figure 1).”

Q5. Line 472: Consider rephrasing PAH "twisted" pathways

R5. We changed the title of the figure 3 accordingly: “Figure 3. Dysregulated pathways governing liver hepcidin expression in PAH.”

Reviewer 2 Report

Thank your for the opportunity to review the work of Quatredeniers et al. They conudected a review evaluating PAH and iron deficiency. The review is nicely written, however, improvements can be made.

1) The definition of iron deficiency - as the authors state - is not straightforward, especially in patients with inflammation (such as PAH patients?). Therefore, the authors should try to postulate a definition of iron deficiency, maybe in a Figure - this would increase the value of this work for the clinician reader.

2) There are many studies ongoing in the field - a table would increase the readability and increase the value of this review.

3) The article lacks a Methods section and the review was not registered (PROSPERO, etc). While a narrative review is OK, the authors must describe their search strategy. Also, for the proposed table (see above), a formal search should be conducted, to asure the reader that the list of studies is complete.

4) Some minor typos in throughout the manuscript (e.g. line 292) should be corrected

Author Response

Q1. The definition of iron deficiency - as the authors state - is not straightforward, especially in patients with inflammation (such as PAH patients?). Therefore, the authors should try to postulate a definition of iron deficiency, maybe in a Figure - this would increase the value of this work for the clinician reader.

R1. A definition of iron deficiency that takes into account inflammation is available in the revised manuscript (reference #68) “According to a recent retrospective study, the ratio of sTfR/(log(serum Ferritin)) (the sTFRF index) corrected for CRP level, might be a better prognostic biomarker to assess iron deficiency in PAH patients, and future studies may consider this issue. Iron deficiency is defined as a sTFRF index > 3.2 if CRP < 0.5 mg/dL, or a sTFRF index > 2 if CRP > 0.5 mg/dL [68].” (C-reactive protein(CRP) being a marker for inflammation).

Q2. There are many studies ongoing in the field - a table would increase the readability and increase the value of this review.

R2. We added table 1 accordingly in the revised manuscript.

Q3. The article lacks a Methods section and the review was not registered (PROSPERO, etc). While a narrative review is OK, the authors must describe their search strategy. Also, for the proposed table (see above), a formal search should be conducted, to asure the reader that the list of studies is complete.

R3. This review is an expert opinion and not a meta-analysis like PROSPERO (PMID: 24717371). For this reason, this article is not registered and methods not detailed.

Q4. Some minor typos in throughout the manuscript (e.g. line 292) should be corrected.

R4. The revised manuscript has been edited accordingly.